# T cell receptor repertoires of mice and humans are clustered in similarity networks around conserved public CDR3 sequences

Asaf Madi[1†], Asaf Poran[1†], Eric Shifrut[1], Shlomit Reich-Zeliger[1], Erez Greenstein[1], Irena Zaretsky[1], Tomer Arnon[1,2], Francois Van Laethem[3], Alfred Singer[3], Jinghua Lu[4], Peter D Sun[4], Irun R Cohen[1], Nir Friedman[1*]

[1]Department of Immunology, Weizmann Institute of Science, Rehovot, Israel; [2]Department of Physics and Astronomy, Alfred University, Alfred, United States; [3]Experimental Immunology Branch, National Cancer Institute, Bethesda, United States; [4]Structural Immunology Section, Laboratory of Immunogenetics, National Institute of Allergy and Infectious Diseases, Rockville, United States

**Abstract** Diversity of T cell receptor (TCR) repertoires, generated by somatic DNA rearrangements, is central to immune system function. However, the level of sequence similarity of TCR repertoires within and between species has not been characterized. Using network analysis of high-throughput TCR sequencing data, we found that abundant CDR3-TCR$\beta$ sequences were clustered within networks generated by sequence similarity. We discovered a substantial number of public CDR3-TCR$\beta$ segments that were identical in mice and humans. These conserved public sequences were central within TCR sequence-similarity networks. Annotated TCR sequences, previously associated with self-specificities such as autoimmunity and cancer, were linked to network clusters. Mechanistically, CDR3 networks were promoted by MHC-mediated selection, and were reduced following immunization, immune checkpoint blockade or aging. Our findings provide a new view of T cell repertoire organization and physiology, and suggest that the immune system distributes its TCR sequences unevenly, attending to specific foci of reactivity.

*For correspondence: nir. friedman@weizmann.ac.il

†These authors contributed equally to this work

Competing interests: The authors declare that no competing interests exist.

## Introduction

The T-cell receptor (TCR), which is generated through random rearrangement of genomic V-D-J segments, is the mediator of specific antigen recognition by T lymphocytes. The collective variety of these receptors expressed by an individual, the TCR repertoire, reflects the state of the adaptive immune system and its history, as its composition changes throughout life in response to immune challenges. The individual TCR repertoire is shaped by biases in the process of VDJ recombination (*Robins et al., 2010*; *Miles et al., 2011*; *Murugan et al., 2012*; *Ndifon et al., 2012*), and by the subsequent expansion and deletion of certain T cell clones upon antigen recognition during T cell development in the thymus, and later in the periphery.

Here, we studied the organization of TCR repertoires using high-throughput TCR sequencing, comparing data from mice and humans. We focused on the CDR3 (complementary determining region 3) amino acid (AA) sequence of the TCR$\beta$ chain, which is the most diverse segment of the TCR and is positioned to interact with the antigenic peptide epitope presented by an MHC molecule (*Davis and Bjorkman, 1988*). The organization of TCR repertoires of individual mice and humans

was evaluated using network analysis, where CDR3 sequences were connected based on their level of sequence similarity.

## Results

Initially, we constructed TCR networks from a dataset of TCR$\beta$ AA sequences obtained from splenic CD4+ T cells from 12 healthy C57BL/6 mice (*Madi et al., 2014*). We obtained on average about 30,000 different CDR3 sequences from each mouse, which were found at varying abundances and had an average length of 13.4 ± 1.4 (mean ±SD) AA. *Figure 1A* shows a network obtained using the thousand most frequent CDR3 sequences from a single mouse, which in terms of abundance correspond to 34% of the total sequences obtained for that mouse. CDR3 sequences (nodes) were connected (by edges) if they were separated by one amino acid difference (replacement/addition/deletion of one AA) – a Levenshtein distance of 1(*Levenshtein, 1966*). A cluster was defined as a set of two or more nodes that are connected to each other by any number of edges and intermediate nodes (*Figure 1A*, inset). A similar analysis had previously revealed the existence of networks of B-cell immunoglobulin heavy-chains, which were attributed to clonally derived sequences generated by somatic hyper-mutations (SHM) (*Ben-Hamo and Efroni, 2011*; *Bashford-Rogers et al., 2013*). Our analysis demonstrated the existence of networks also for TCR$\beta$ sequences. As T cells do not undergo SHM, other factors lead to the formation of TCR similarity networks.

We repeated this analysis for all 12 mice, and found that of the thousand most frequent CDR3 sequences in each mouse (with an accumulated frequency of 34.5 ± 8% of total sequences), 647 ± 104 (mean ±SD) were clustered, with 1282 ± 383 edges. In contrast, networks composed of a thousand randomly selected CDR3 sequences from a single mouse (with an accumulated frequency of 5 ± 0.7% of total sequences) were much sparser (*Figure 1B*), with only 225 ± 64 sequences clustered, and with 152 ± 52 edges (average values for 10 independent randomized sets of sequences). These results were not sensitive to the number of sequences used for the analysis (*Figure 1—figure supplement 1*).

To contrast the TCR networks with their BCR counterparts, we tested whether these networks are structurally similar. BCR networks have been shown to center around highly abundant clones, representing a snapshot of the individual-specific local evolution driven by SHM. However, we found no correlation ($R^2 = 0.11 ± 0.07$) between the abundance of a TCR CDR3 sequence and its degree of connectivity in the network (number of edges connecting it to other sequences). We further found that each cluster typically contained sequences of a single (or in some cases two) specific J segment (*Figure 1—figure supplement 2*). V usage, in contrast, was not cluster-specific; any cluster contained sequences with many different V segments (*Figure 1—figure supplement 2*). This reflects the higher number of V segments compared with J segments, as well as their lower overlap with CDR3 and the relative similarity of their 3' ends. Networks of similar connectivity were obtained also for the top 1000 CDR3$\beta$ sequences from CD8 T cells, and for CD4 T cells of a different mouse strain (C3H.HeSnJ), that bears a different MHC haplotype (H2$^k$; *Figure 1—figure supplement 3*, *Figure 1—figure supplement 4*).

We found a parallel network organization also in human TCR$\beta$ repertoires: we analyzed previously published data containing the TCR$\beta$ repertoires of 39 human subjects of different ages (*Britanova et al., 2014*), and found that the most abundant CDR3 sequences formed connected clusters in human TCR repertoires (*Figure 1C*, *Supplementary file 1*, and *Figure 1—figure supplement 1*), though with a lower connectivity than that found in the similarity networks of inbred mice. From the thousand most frequent CDR3 sequences (accumulated frequency of 17.1 ± 6.6% of total sequences) in each of the 11 young human subjects in that study (ages 6–25 years), 207 ± 79 nodes were clustered, with 367 ± 201 edges. Networks composed of randomly selected sequences from the individual subjects generated only 8 ± 4 clustered nodes with 4 ± 2 edges. We thus conclude that these newly discovered TCR similarity networks are likely to be driven by conserved evolutionary forces, as opposed to BCR networks that are generated by SHM that operates within individuals.

Next, we tested whether these TCR networks reflect our previous finding that TCR$\beta$ CDR3 AA sequences express a range of sharing levels between individual mice. As a measure of sharing level, we used a reference dataset of 28 mice (*Madi et al., 2014*) and assigned to each CDR3 AA sequence in a network a sharing level ranging from 1 (private, found in only one mouse in the reference dataset) to 28 (public, found in all 28 mice in the reference dataset) (*Madi et al., 2014*).

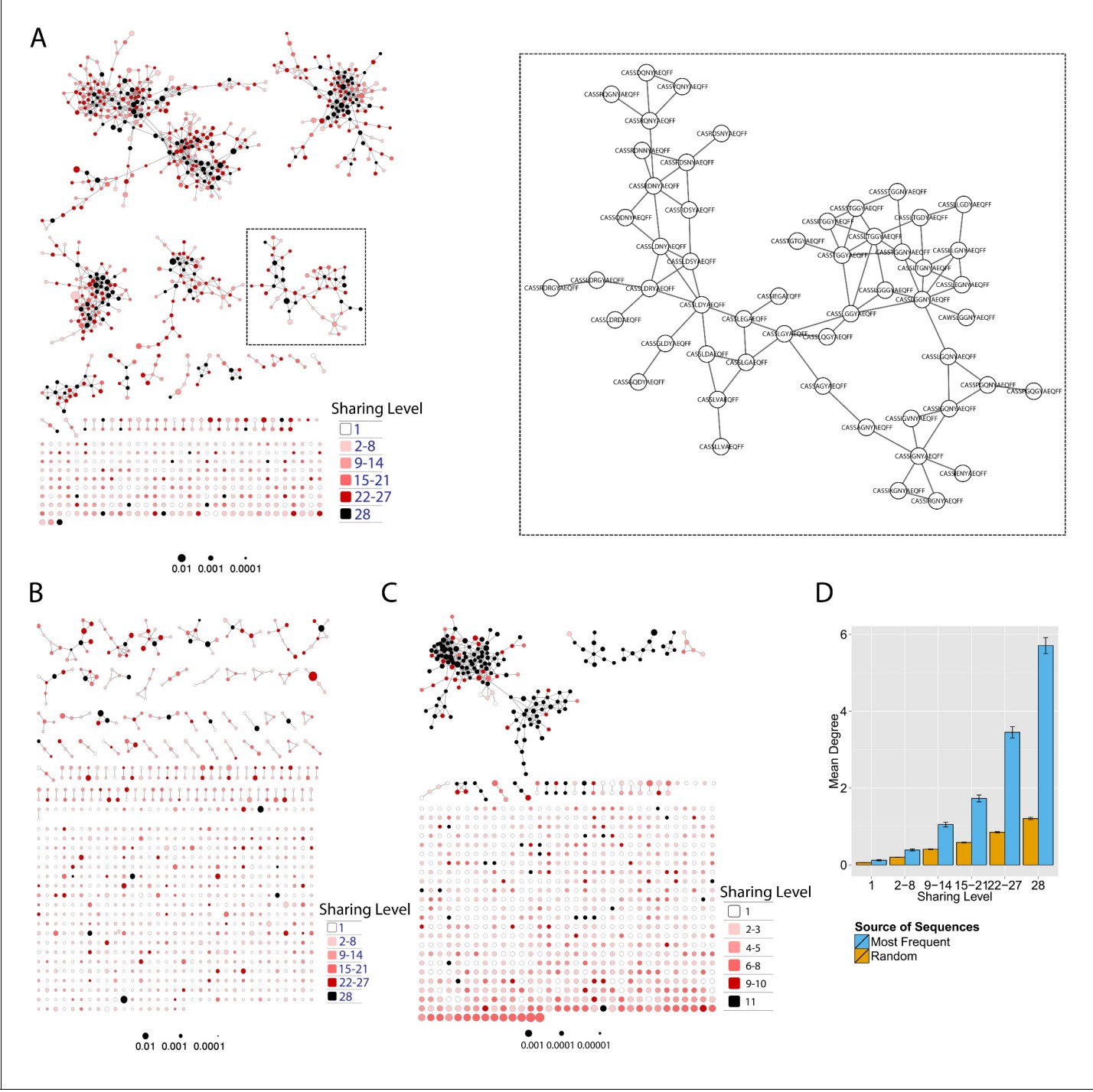

**Figure 1.** Mouse and human TCR repertoires manifest dense similarity networks surrounding public CDR3β sequences. (**A**) Networks formed by the thousand most frequent CDR3 AA sequences expressed in the TCRβ repertoire of splenic CD4 T cells from a single mouse. Nodes (CDR3 AA sequences) were connected by edges defined by a Levenshtein distance of 1 (one AA substitution/insertion/ deletion). Node size reflects its log frequency (scale at the bottom). The nodes are colored according to their sharing levels in a reference dataset of 28 mice (*Madi et al., 2014*), from *Private* CDR3 sequences (white, found in only one mouse in the reference dataset) to *public* (black, shared by all 28 mice). Inset shows a blowup of the marked cluster with labeled CDR3β AA sequences (nodes) and edges which represent a Levenshtein distance of 1 between connected nodes. (**B**) Networks formed by a thousand CDR3β sequences randomly chosen from the repertoire of a single mouse. (**C**) A Network formed by the thousand most frequent CDR3 AA sequences in the TCRβ repertoire of a representative human subject (data from [*Britanova et al., 2014*]). Nodes are colored by their degree of sharing among the 11 young subjects in that study (ages 6–25 years). (**D**) Mean degree of node connectivity as a function of sharing

*Figure 1 continued on next page*

*Figure 1 continued*

level in a network formed by the top 1000 CDR3 sequences (blue) or by 1000 randomly chosen sequences (orange). Error bars indicate standard error (SE) across the 12 mice used in this study.

The following figure supplements are available for figure 1:

**Figure supplement 1.** Mean number of clustered nodes as a function of the sample size selected for generating the network.

**Figure supplement 2.** CDR3$\beta$ sequences form networks with clusters dominated by J-genes and heterogeneous for V-genes.

**Figure supplement 3.** CD8$^+$ T cell networks formed by the thousand most frequent CDR3 AA sequences expressed in two mice.

**Figure supplement 4.** Networks from C3H.HeSnJ mouse strain bearing the H2$^k$ MHC haplotype.

**Figure supplement 5.** Evaluating the level of node centrality vs. sharing level.

**Figure supplement 6.** Node centrality vs. sharing level in human TCR$\beta$ repertoires.

Interestingly, we found a strong association between the sharing level of a CDR3 sequence and its connectivity in the network: highly shared sequences are positioned at the center of network clusters (*Figure 1A*). This is indicated by a statistically significant correlation between the degree of node connectivity (number of edges connecting it to other nodes in the network) and its sharing level (*Figure 1D*), (R = 0.69 ± 0.03, p-value<2.2e-16; see also *Supplementary file 1*). An independent method for estimation of node centrality, betweenness centrality, confirmed the correlation between CDR3 sharing and centrality for the 1000 most abundant CDR3 sequences, but not for a random set of expressed sequences (*Figure 1—figure supplement 5*, *Supplementary file 1*). As in mice, public CDR3 sequences in humans manifested a higher degree of connectivity than did more private sequences (*Figure 1C*, *Figure 1—figure supplement 6*), and sequence abundance was not correlated with its level of connectivity (*Supplementary file 1*). Thus, private and public CDR3 sequences are distributed differently across the mouse and human networks: public sequences are highly connected to other similar sequences and are more central in network clusters; in contrast, more private sequences are found at the edges of clusters, or as un-connected nodes, with rare similarity to other sequences in the network.

These findings of a similar organization of mouse and human TCR networks prompted us to look for the existence of shared CDR3$\beta$ sequences between the two species. Interestingly, we found that a substantial number of TCR$\beta$ CDR3 AA sequences were shared by mice and humans. Out of 5,247,785 unique AA sequences in the human dataset (11 young individuals) and 371,977 in the mouse dataset (28 animals), 27,337 were shared by at least one mouse and one human individual. In general, CDR3 sequences with a higher level of sharing in mice were found to have an increased probability of being found in human repertoires; similarly, sequences more shared in humans were found more frequently in mice (*Figure 2A*, *Figure 2—figure supplement 1*). Of note, more than 25% of the public CDR3 sequences (found in all 11 young human subjects, or found in all 28 mice) were found also in at least one individual of the other species (*Figure 2A*).

We defined a set of cross-species (CS) public CDR3 sequences that were public or relatively public in both mice (found in at least 25 of the 28 mice) and humans (found in all 11 young individuals). All these 86 CS-public sequences contained the human J$\beta$2.7 or J$\beta$2.3 segments, and the mouse J$\beta$2.5 or J$\beta$2.7 segments. V usage was dominated by V$\beta$20.1 in humans, but a more diverse V usage was observed in mice. Examples of CS-public sequences are shown in *Figure 2B*. The CS-public CDR3 sequences manifested a significantly higher degree of connectivity in human and mouse networks than did CDR3 sequences that were public only in humans, only in mice or not public in either (*Figure 2C,D* and *Figure 2—figure supplement 2*). Moreover, we found a significant correlation between the mean degrees of CS-public sequences in mouse and human networks (*Figure 2—figure supplement 3*); CS-public sequences that have more neighbors in mouse networks also tended to have more neighbors in human networks, suggesting an evolutionarily conserved network structure. We note that while CS-public sequences are central in network clusters, their frequency is not

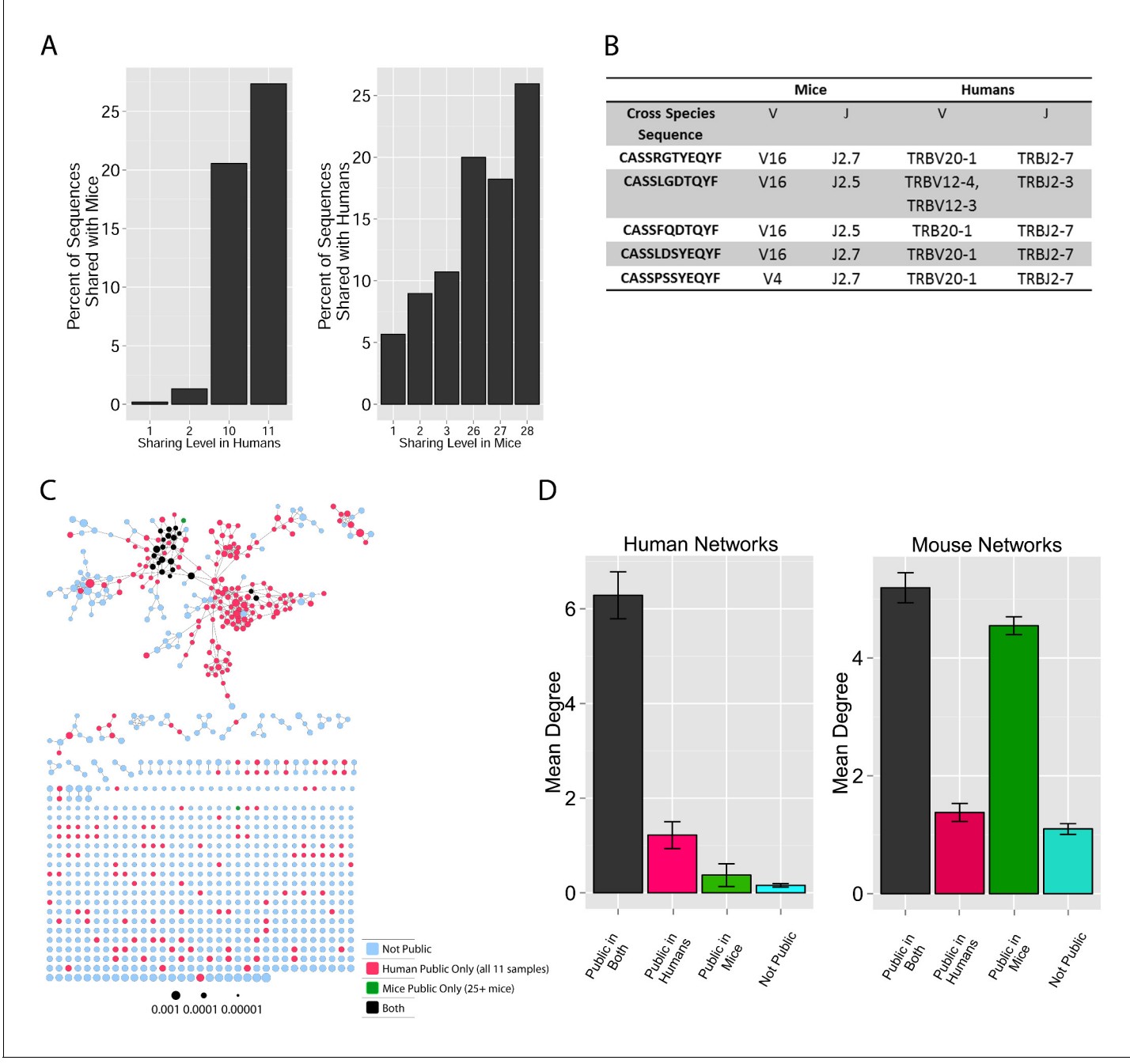

**Figure 2.** TCR repertoires are focused around public and cross species- (CS-) public CDR3 AA sequences shared by mice and humans. (**A**) Human (left) or mouse (right) CDR3 sequences are grouped according to their sharing level in the corresponding dataset. For each sharing group, we plotted the percentage of sequences that were shared by at least one subject of the other species. (**B**) Examples of CS-Public CDR3 sequences, and their V and J segments in mouse and human repertoires. (**C**) A network formed by the top 1000 CDR3 sequences of a single human subject. Node color represents its sharing within or between species: Pink - shared by all 11 human subjects; Green - shared by at least 25 of the 28 mice; Black – CS-public nodes shared by all 11 humans and at least 25 mice; Blue - not shared. (**D**) The mean number of edges per node (degree) in the 11 human and 28 mouse networks, subdivided into the four categories as in C. Error bars mark SE.

The following figure supplements are available for figure 2:

**Figure supplement 1.** Cross-species TCR sharing.

**Figure supplement 2.** Sharing properties of the 86 observed CS-public CDR3 sequences in simulated data.

*Figure 2 continued on next page*

*Figure 2 continued*

**Figure supplement 3.** CS-Public CDR3 sequences are central in mouse TCR*β* networks.
**Figure supplement 4.** Degree of CS-public sequences is correlated in mouse and human TCR networks.

higher than that of other public sequences that are found only in humans or in mice. These findings propose that similar driving forces may generate and expand particular public CDR3 TCR sequences that contain conserved sequence motifs in the two species.

To further characterize the mechanisms that contribute to the generation of CS-public sequences, we evaluated their existence in synthetic TCR repertoires that simulate the random generation of TCR sequences (see methods). These simulations do not include any clonal selection, thus they allow discrimination between genetic mechanisms that influence the generation of TCRs and selection mechanisms that shape it somatically. We generated 100 datasets of simulated repertoires of 28 mice and 11 humans, the sizes of which matched the sizes of the experimental repertoires. The simulated repertoires contained a somewhat larger number of CS-public CDR3 sequences than observed in the experimental data (average of 221 ± 9 in the simulations, vs. 86 in the data). The simulated CS-public sequences contained the same restricted set of mouse and human J segments, which are highly similar between the two species (J2.7 mouse and human; J2.5 mouse/J2.3 human). Thus, sequence homology of J segments contributes to the formation of CS-public TCRs, but is not sufficient by itself, and is accompanied by other mechanisms that induce bias in the recombination process (e.g. biased V segment usage, statistics of nucleotide deletions and insertions at V-D and D-J junctions). We also asked whether the simulated repertoires contained the same CS-public sequences as those observed experimentally. We found that 54 out of the 86 experimentally observed CS-public sequences were identical to simulated CS-public sequences, while 32 were not CS-public in the simulations (*Figure 2—figure supplement 4*). The partial overlap between simulations and data may result from inaccuracies in the assumptions of the simulations regarding the random TCR generation process, or indicate that selection mechanisms in the thymus and in the periphery further influence the existence of specific CS-public sequences.

We further evaluated the similarity between public sequences by analyzing the level of connectivity within a network composed of the most highly shared CDR3 sequences. A network formed by the 1000 most public mouse sequences (found in >25 of the 28 mice) was highly connected, with 965 clustered nodes and 3387 edges (*Figure 3A*). In contrast, networks formed by the 1000 most abundant *private* sequences (found in only one of the 28 mice) were very sparse, manifesting only 38 ± 15 clustered nodes and 20 ± 7 edges (mean ± SD, averaged over 28 mice). Similarly, a network formed by the 1000 most public human CDR3 sequences was also highly connected (with 969 clustered nodes and 4398 edges, *Figure 3B*).

The functional TCR is formed by a complex of TCR alpha and beta chains (*Davis and Bjorkman, 1988*), hence one cannot attribute specific antigen recognition to CDR3*β* segments alone. Moreover, the current level of understanding precludes the development of general predicting tools that can computationally relate a TCR sequence to an antigen that it recognizes. Defining TCR antigen specificity is further complicated by substantial TCR cross-reactivity (*Burrows et al., 1997*; *Wooldridge et al., 2012*). Yet, TCR*β* sequences that bind the same pMHC antigen do contain shared CDR3*β* sequence motifs (*Klinger et al., 2015*; *Chen et al., 2017*; *Sun et al., 2017*; *Tickotsky et al., 2017*). Thus, some insight on antigen specificity can be gained by linking the sequence-similarity networks to previously annotated TCR sequences. We have reported that 124 of the CDR3*β* sequences in our mouse dataset were associated with various mouse immune reactivities previously described in the literature (*Madi et al., 2014*). As a step towards relating antigen specificity to the clusters of public CDR3 sequences, we looked for these 124 annotated CDR3*β* sequences within the clusters of shared CDR3 sequences. The annotated sequences were grouped according to four categories: a) Immunity to foreign pathogens; b) Allograft reactions; c) Tumor-associated T cells; and d) Autoimmune conditions. *Figure 3A* includes these annotations in the network formed by the 1000 most public CDR3*β* sequences. Out of the 124 annotated sequences, 63 were either identical to one of the existing nodes (n = 11), or linked to an existing node by a Levenshtein distance of 1

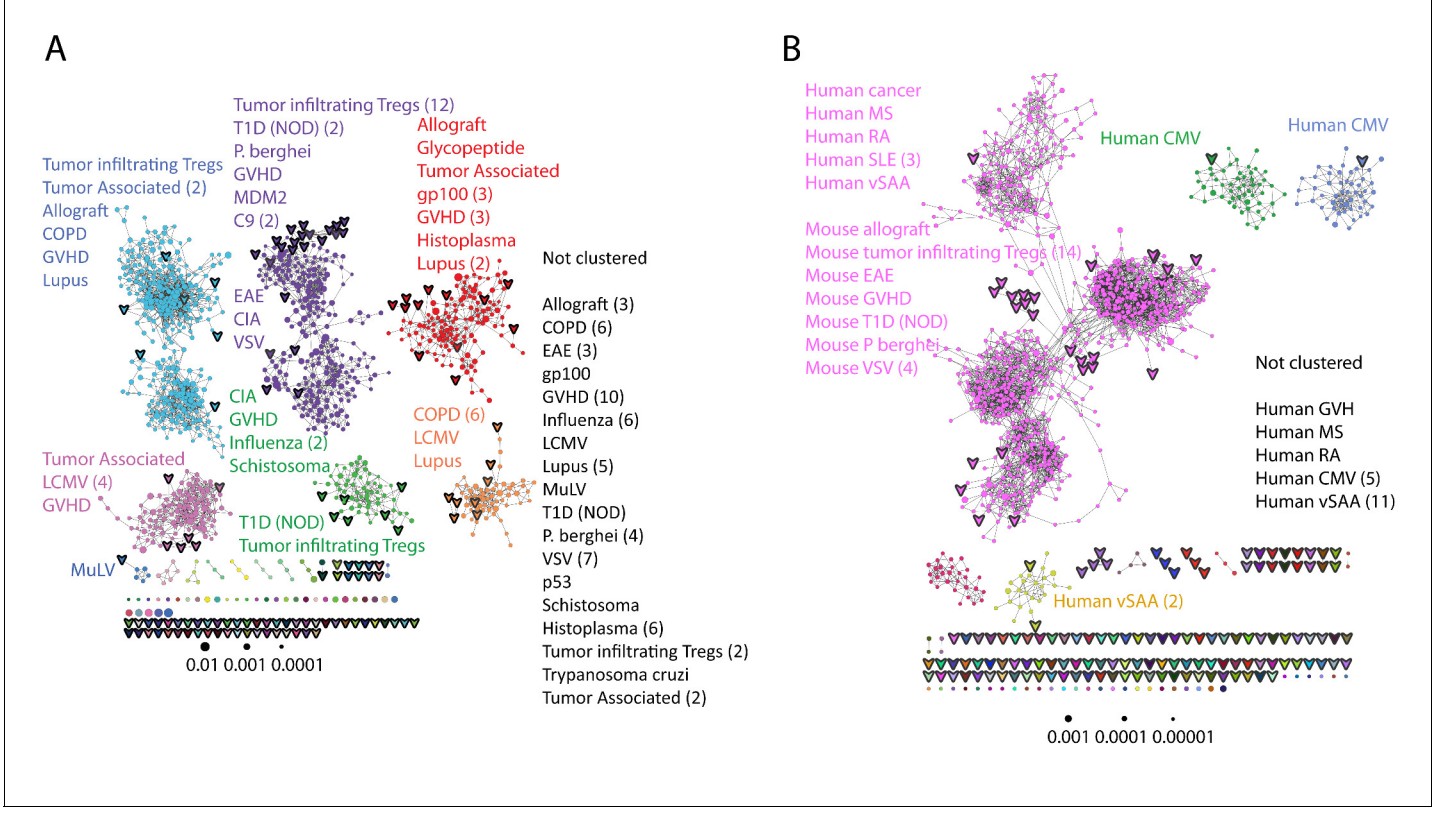

**Figure 3.** Public CDR3 sequences form highly connected similarity networks in mice and humans and are enriched for self-associated immune reactivities. (**A**) A network formed by the 1000 most shared mouse CDR3 sequences (found in >25 of 28 mice). Node size corresponds to the mean abundance of the sequence. Nodes are colored according to their cluster association. 124 CDR3 sequences that were previously annotated (see [**Madi et al., 2014**]) were added to the network and are presented as arrowheads. 63 annotated sequences were either identical to, or at a Levenshtein distance of 1 from one of the nodes, and are listed next to each cluster (with the corresponding color). Annotations of 61 un-clustered sequences are also listed. (**B**) A network formed by the 1000 most frequent public CDR3 sequences in humans (found in all 11 subjects). Previously annotated mouse (n = 124) and human (n = 30) CDR3 sequences were added to the network as in A (arrowheads). The clusters were distinctly colored in order to visually match between clusters and their annotated sequences, not to define antigen specificity of a cluster. A list of linked annotated CDR3 sequences is shown next to each cluster (11 of 30 human and 23 of 124 mouse annotated CDR3 sequences), together with a list of unclustered annotated human sequences.

The following figure supplement is available for figure 3:

**Figure supplement 1.** Public CDR3 sequences form highly connected similarity networks in mice and are enriched for self-associated immune reactivities.

(n = 52). The clustered annotated nodes were found to be enriched with annotations related to self or self-like autoimmune, cancer or allograft reactions (self-related: 51/63 = 81% of network-clustered sequences vs. 85/124 = 69% in all 124 annotated sequences, compared to non-self: 12/63 = 19% in clusters vs. 39/124 = 31%; Fisher exact test p=0.0035).

We find that sequences with a similar annotation tended to be linked in the same cluster. Examples include twelve sequences of tumor infiltrating regulatory T cells (*Sainz-Perez et al., 2012*) which were found in cluster #2; six COPD related CDR3 sequences (*Motz et al., 2008*) in cluster #6; and four CDR3 sequences connected with cluster #2 that were associated with type 1 diabetes in NOD mice in two different studies (*Nakano et al., 1991*; *Tikochinski et al., 1999*). However, different annotations can also be found in the same cluster (*Figure 3A*); for example, mouse CDR3 sequences associated with experimental autoimmune encephalomyelitis (EAE; [*Menezes et al., 2007*]) and collagen-induced arthritis (CIA; [*Osman et al., 1993*]) were also connected to cluster #2. *Figure 3B* shows that many previously annotated self/self-like sequences of humans and mice were also linked

to clusters in the network of public human sequences. Thus, the CDR3 clusters, which serve as repertoire foci, seem to be enriched with TCR sequences that are associated with self (or self-like) reactivities, whereas pathogen-associated TCR sequences are less clustered and so tend to be more evenly spread throughout sequence space.

To analyze mechanisms involved in network formation, we investigated the contribution of antigen selection using two complimentary approaches. First, we analyzed similarity networks formed by CDR3 sequences of CD4$^-$CD8$^-$double-negative (DN) thymocytes. Rearranged TCR$\beta$ chains in DN cells are not subject to MHC-dependent selection, which only occurs at later stages of thymic development. We found that networks formed by DN CDR3 sequences were significantly less connected compared to splenic CD4$^+$ T cells, which have undergone antigen selection (*Figure 4A* and *Supplementary file 2*). In addition, DN thymocytes and CD4$^+$ spleen T cells manifested different levels of convergent recombination (*Venturi et al., 2006*, *2008*). Public CDR3 AA sequences in DN thymocytes were encoded on average by a low number of nucleotide (nt) sequences, whereas the same AA sequences were encoded by a much larger number of nt sequences in CD4$^+$ splenic T cells (*Figure 4C*, *Figure 4—figure supplement 1*). The finding of relatively increased network clusters in T cells that have undergone antigen selection suggests that the CDR3 AA sequences that are found within clusters are positively selected; this antigen selection would extend any underlying physical bias generated during TCR DNA recombination in the thymus (*Murugan et al., 2012*; *Ndifon et al., 2012*).

To further study the impact of selection, we evaluated TCR networks formed in the repertoires of splenic T cells from mice lacking four elements needed for physiological MHC-dependent antigen selection: MHC-I and -II molecules together with CD4 and CD8 co-receptor molecules, so-called Quad-KO mice (*Van Laethem et al., 2007*, *2013*). In contrast to wild-type (WT) mice, the TCR of Quad-KO mice are selected by MHC-independent ligands in the thymus and their T cells express a diverse MHC-independent TCR repertoire in the periphery (*Van Laethem et al., 2007*; *Tikhonova et al., 2012*; *Van Laethem et al., 2013*). We found that similarity networks formed by the top 1000 CDR3 sequences from Quad-KO mice were significantly less connected than those of the WT strain (C57BL/6) measured in the same set of experiments (*Figure 4A* and *Supplementary file 2*). Together, these findings indicate that MHC-dependent thymic selection plays a significant role in promoting the formation of dense clusters of TCR-similarity networks. Lack of MHC-dependent selection in DN thymocytes and in Quad-KO mice is associated with TCR networks of reduced connectivity; in contrast, TCRs that are subject to MHC selection form dense networks with a higher level of convergent recombination. Thus, recombination biases combined with clonal selection generate a TCR repertoire that is not uniform, but rather focused in specific regions of sequence space that are preferentially associated with self-related antigen-reactivities.

Following these observations, we tested if the relative abundance of CS-public clonotypes is increased by MHC-dependent selection. To this end, we compared the frequency of CS-public sequences in repertoires of Quad-KO mice and DN thymocytes to those of control WT mice (*Figure 4B*). The cumulative frequencies of the CS-public CDR3 sequences between two sets of experiments done with WT mice (the 28 WT mice used in the network analysis, and the WT mice used as controls in the Quad-KO experiment) show no significant difference (P value = 0.293). On the other hand, the Quad-KO repertoires exhibited lower total frequency of the CS-public CDR3s compared with both 28 WT mice (P value = 4.318e-09) and the Quad-WT mice (P value = 0.01781). The cumulative frequency in the DN shows a similar trend, with no statistical significant (P value = 0.1877). Together, these results indicate that, although sequence homology of V and J germline segments between mice and humans and bias in the recombination process influence the probability for a sequence to be shared between the two species, additional selection forces are influencing its abundance.

Since the composition of the TCR repertoire of an individual changes in response to immune challenges throughout life, we tested the effects of both immunization and aging on the network organization of the TCR repertoire. We immunized naïve mice with p277, a self peptide derived from HSP60 (heat shock protein 60), or with a foreign peptide, derived from ovalbumin (OVA). Peptide p277 was previously found to be recognized by the C9 public TCR in NOD mice (*Tikochinski et al., 1999*), and the CDR3$\beta$ sequence of the C9 clone was also public in C57BL/6 mice (*Madi et al., 2014*). Additionally, we analyzed the network structures in the TCR repertoires of T cells from the immunized mice that were further cultured in vitro with antigen presenting cells loaded with the

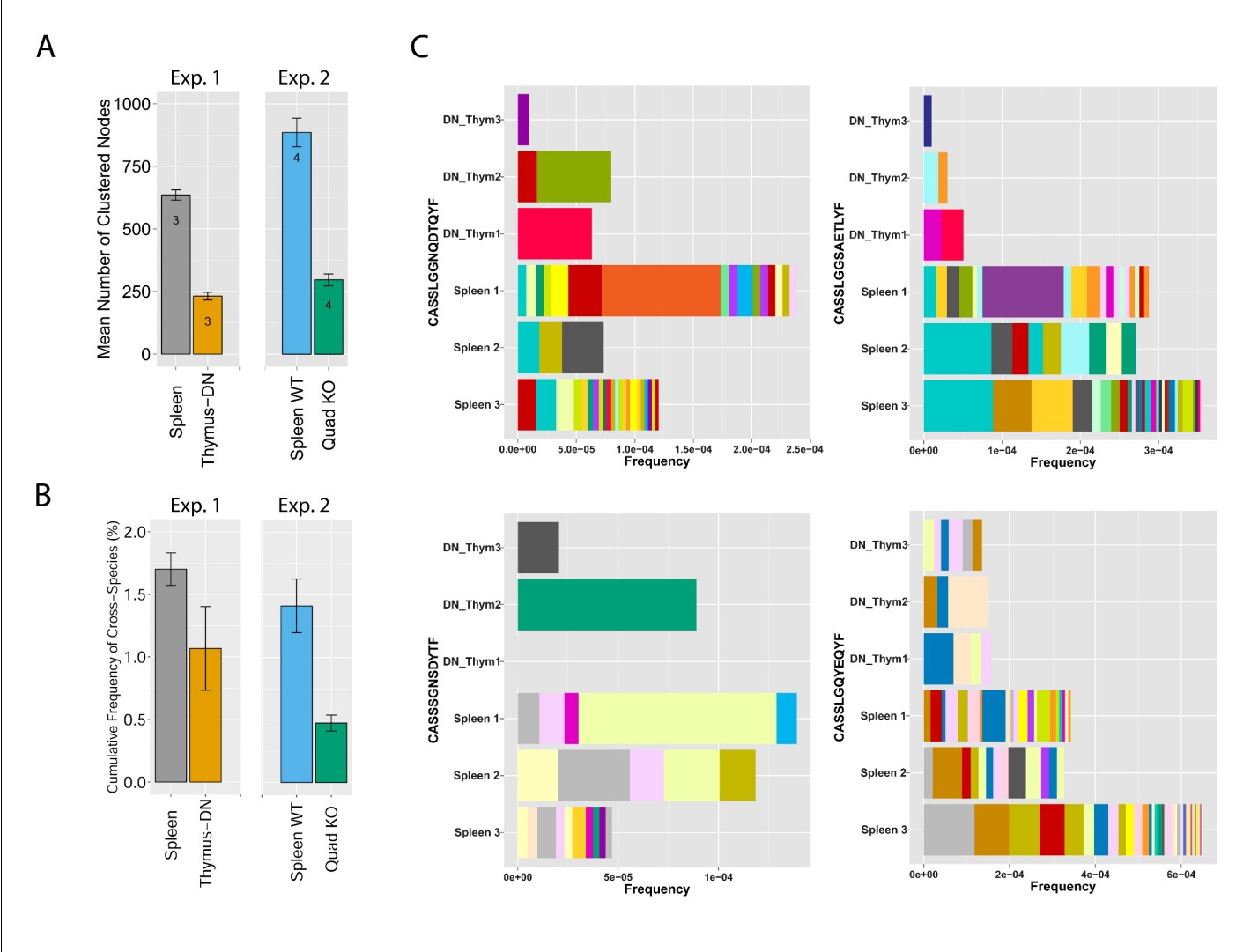

**Figure 4.** MHC-dependent public CDR3 sequences form highly connected similarity networks. (**A**) Mean number of clustered nodes in networks formed by the top 1000 CDR3 sequences from the following repertoires: DN thymocytes (CD4⁻CD8⁻) (n = 3), CD4⁺ spleen T cells (n = 3), Quad-KO mice (**Van Laethem et al., 2007**) (lack MHC-I, MHC–II, CD4 and CD8) (n = 4), and their WT controls (C57BL/6) (n = 4). Error bars signify standard error. (**B**) Cumulative frequency of the 86 CS-public CDR3 sequences (observed in the reference datasets of 28 WT mice and 11 healthy humans) is shown for: DN thymocytes (CD4⁻CD8⁻) (n = 3), CD4⁺ spleen T cells (n = 3) (left), Quad-KO mice (n = 4), and their WT controls (C57BL/6) (n = 4). Error bars signify standard error. (**C**) Cumulative frequency of nucleotide sequences coding for two annotated (C9 and COPD, top) and two unknown (bottom) public AA CDR3 sequences from repertoires of DN thymocytes and CD4⁺ spleen T cells (sequences from 3 mice are shown). Each color represents a different nucleotide sequence.

The following figure supplement is available for figure 4:

**Figure supplement 1.** DN thymocytes manifest lower convergent recombination.

specific peptide. The distribution of sequence abundances and repertoire evenness were evaluated using the Gini inequality coefficient, which ranges from 0 for a repertoire where every sequence is present in equal abundance, to 1 for a repertoire dominated by a single sequence, with other sequences present at zero abundance (**Bashford-Rogers et al., 2013**; **Thomas et al., 2013**).

We found that immunization with either peptide resulted in repertoires that contained a set of expanded CDR3 sequences and had an increased abundance inequality. In vitro re-stimulation further increased inequality (**Figure 5A–C** and **Supplementary file 3**). This inequality was associated

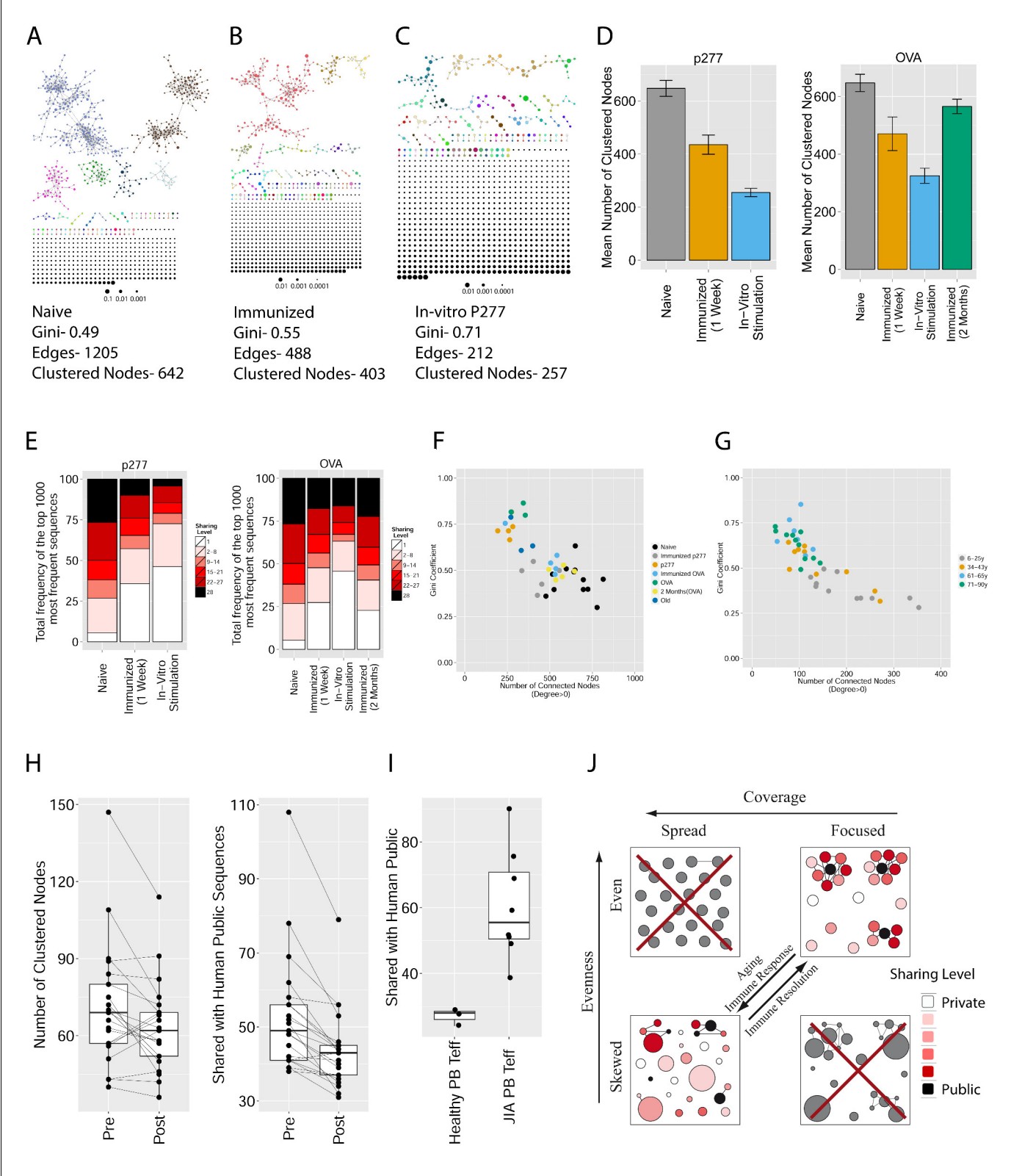

**Figure 5.** Immunization, in vitro antigen re-stimulation, anti-CTLA4 antibody treatment and aging perturb TCR networks coupled with an increase in repertoire skewness. (A–C) Networks of the thousand most frequent CDR3 sequences are shown for (A) a naïve mouse, (B) a mouse Immunized with a self-peptide (p277), and (C) T cells from the spleen of an immunized mouse, which were re-stimulated in vitro with the p277 peptide. (D) Mean number of clustered nodes in networks formed by the top 1000 CDR3 sequences from the following repertoires: Left: naïve mice (n = 12); p277 immunized mice,

*Figure 5 continued on next page*

*Figure 5 continued*

7d post immunization (n = 5); and in-vitro re-stimulated with p277 (n = 5). Right: naïve mice (n = 12); OVA immunized mice, 7d post immunization (n = 5); in-vitro re-stimulated with OVA peptide (n = 3); and immunized mice, 2 months post-immunization (n = 5). Error bars indicate standard error. (E) Frequency of the top 1000 most frequent CDR3 sequences by sharing level, for the same repertoires as in (D). Sharing levels were calculated based on sharing in the reference dataset of 28 mice. (F) The Gini Coefficient (a measure for repertoire evenness) plotted vs. the number of clustered nodes, for the top 1000 CDR3 sequences from the repertoires from (D, E) and from aged mice (n = 3). (G) The Gini Coefficient plotted vs. the number of clustered nodes for 39 human samples (*Britanova et al., 2014*) divided into 4 age groups. (H) The number of clustered nodes (left) and the number of public clonotypes (right, shared by all 11 young human samples in a reference cohort [*Britanova et al., 2014*]) for the top 1000 most abundant CDR3 sequences in 21 paired samples of patients at baseline and 30 to 60 days after receiving CTLA4 blockade treatment with tremelimumab (data from [*Robert et al., 2014*]). (I) Number of public clonotypes (defined as in H) out of the top 1000 most abundant CDR3 sequences in either healthy donors (left) or Juvenile Idiopathic Arthritis patients (right). (J) A conceptual figure of the evolution of repertoire structure. In young and healthy individuals the repertoire is focused and even (top-right), with public and CS-public CDR3 sequences at the center of network clusters. Following an immune response, or with aging, the repertoire becomes more skewed and spread in sequence space (bottom-left), due to preferential expansion of private clones at the expense of more public clones.

The following figure supplements are available for figure 5:

**Figure supplement 1.** Immunization and in vitro antigen stimulation affect network architecture.

**Figure supplement 2.** Mouse TCR Networks become less connected with aging.

**Figure supplement 3.** Human TCR Networks become less connected with aging.

**Figure supplement 4.** With aging, the repertoire becomes more skewed and spread in sequence space due to preferential expansion of private clones at the expense of more public clones.

**Figure supplement 5.** CTLA4 blockade results in a repertoire that is more skewed and spread in sequence space, due to preferential expansion of private clones at the expense of more public clones.

with the emergence of private clones that dominated the post-immunization repertoire, such that the relative weight of public clones was reduced (*Figure 5E*). Interestingly, immunization was also associated with network disruption; the number of clustered nodes and the number of edges both fell after immunization in vivo and fell further after in vitro re-stimulation (*Figure 5D*, *Figure 5—figure supplement 1*). Both the increased inequality and the decreased network connectivity reversed spontaneously in the OVA-immunized mice 2 months following immunization (*Figure 5D,E* (right), *Figure 5—figure supplement 1*). Similar to immunization, repertoires in aged mice (*Figure 5F*, *Figure 5—figure supplement 2*) and in aged humans (*Figure 5G*, *Figure 5—figure supplement 3*) were more unequal and less connected than those of young individuals, and private CDR3 sequences became relatively more abundant with age (*Figure 5—figure supplement 4*). Altogether, we found a strong anti-correlation between the Gini Coefficient of TCR inequality and the number of connected nodes in TCR networks in mice (*Figure 5F*, Spearman correlation = −0.661) and in humans (*Figure 5G*, Spearman correlation = −0.865).

Another factor that impacted network structure was immune checkpoint blockade. We used published CDR3$\beta$ sequence data (*Robert et al., 2014*) from subjects who had undergone CTLA4 (cytotoxic T–lymphocyte-associated protein 4) blockade with tremelimumab. Previous analysis of these data showed that this treatment diversified the peripheral T-cell pool. Applying TCR similarity network analysis, we now show that the 1000 most abundant CDR3 sequences after check-point blockade are less connected than pre-treatment (p value<0.05 ranked Wilcox paired test, *Figure 5H* left); moreover, this reduction in connectivity was detected concurrently with a decrease in the number of public CDR3 sequences and an increase in the frequency of private ones (p-value=0.01947, ranked Wilcox paired test, *Figure 5H* right, *Figure 5—figure supplement 5*). Thus, broadening of the peripheral repertoire following CTLA4 blockade reduces the presence of public clones and enhances the expansion of private clones, similar to the changes we observed in aging or after immunization. This finding raises the possibility that check-point associated immune regulation also could be involved in the prominence of network connectivity of public T cells. Finally, we analyzed TCR repertoires of patients with the autoimmune disease Juvenile Idiopathic Arthritis (JIA)(*Henderson et al.,*

*2016*). We found that there was a strong increase of public (network promoting) TCRs in the peripheral blood of JIA patients compared to healthy donors (P value = 0.0006, *Figure 5I*). Thus, while immune perturbations such as immunization and aging lead to reduced levels of public clonotypes and a reduction in network connectivity, this specific autoimmune condition is associated with an increased level of public clones which are putatively associated with self-antigens.

## Discussion

Our application of network analysis to TCR$\beta$ CDR3 sequencing data reveals a hitherto unrecognized structure of the TCR repertoire in both mice and humans: In young, healthy individuals, the most abundant TCR$\beta$ CDR3 sequences are distributed unevenly in sequence-space, with clusters centered around public CDR3s, and in particular around CS-public sequences, which are public both in mice and humans (*Figure 5J* top-right, even and focused repertoire). The clustering of the most abundant CDR3 sequences in young and healthy individuals results in a repertoire that is much more restricted than would be expected from the random process of TCR somatic recombination. This basic network architecture is modified by immunization and aging due to the dominant expansion of more private CDR3 clonotypes. Thus, public CDR3s that serve as hubs of the TCR networks become less prominent, leading to reduced connectivity of the TCR networks combined with a more skewed repertoire (*Figure 5J* bottom-left, skewed and spread repertoire). We find that network organization and repertoire evenness are restored with the resolution of immune responses. It might be the case that incomplete resolution of immune responses throughout life lead to accumulation of changes in the TCR repertoire that eventually result in the skewed and spread (less clustered) repertoires that we observe in aged individuals. Interestingly, TCR repertoires from patients with the autoimmune condition JIA showed increased levels of public TCR sequences. This aligns with our observation that public TCR networks are enriched with self-associated TCRs. Taken together, our analysis supports the idea that the level of network connectivity, frequency of public TCRs and repertoire evenness are linked to each other, and are concurrently modulated by the individual's immune state (disease/immunization/ aging).

Mechanistically, we found that MHC-dependent antigen selection contributes to the formation of dense networks, since reduced network connectivity was observed in pre-selection DN thymocytes and also by inhibiting MHC-dependent selection, in the Quad-KO mice. These results can be explained by preferential selection and increased survival, in both the thymus and periphery, of T cells that carry specific CDR3 sequences that recognize self-antigens presented by MHC molecules. Different T cell clones, which carry different CDR3 nt sequences but encode the same AA sequence, would appear to enjoy a common selective advantage and accumulate in the peripheral repertoire. This mechanism can explain our observations of increased convergent recombination in splenic CD4[+] T cells compared to DN thymocytes (*Figure 4—figure supplement 1*). Antigen selection can also account for the enhanced network connectivity of TCRs that differ by one AA in their CDR3 sequences; such related CDR3 sequences may be selected by the same peptide-MHC complex, albeit with different affinities (*Moss et al., 1991*; *Serana et al., 2009*; *Zoete et al., 2013*). This working hypothesis needs to be tested experimentally to see if linked CDR3 sequences really cross-react with the same or similar peptide-MHC complexes. MHC-antigen selection of public CDR3 sequences takes place on a background of biases in the biophysical process of DNA recombination (*Elhanati et al., 2014*). Combined, these processes lead to the formation of dense network clusters of the most abundant public TCR sequences, as we report here. In contrast, the most abundant private TCR sequences generate poorly connected networks. B cell receptor (BCR) sequences (*Ben-Hamo and Efroni, 2011*; *Bashford-Rogers et al., 2013*), unlike the T-cell repertoire networks we disclose here, have long been known to generate networks in individual subjects by affinity maturation that is mediated by SHM; T cells do not undergo SHM so TCR networks must be generated in the developmental process. Thus, dominant and public T cell clonotypes have a higher sequence similarity than non-dominant and private ones. In contrast, BCR networks have a distinct structure resulting from the SHM process, in which abundance and degree are correlated, which is not the case in TCR networks.

Our finding that TCR CDR3 networks include identical and related sequences that are not confined to individuals but are shared by most individuals of the same species and even cross the species divide between mice and humans, suggests the likelihood of some fundamental evolutionary

advantage in such sequences. As noted above, antigen specificity of a TCR cannot be defined based on its CDR3$\beta$ alone. However, the same or very similar CDR3$\beta$ sequences are frequently observed within repertoires of T cells specific for a given antigen, in combination with flexible or preferential pairing with TCR$\alpha$ (*Klinger et al., 2015*; *Chen et al., 2017*; *Tickotsky et al., 2017*). Hence, we hypothesize that T cell clones bearing the conserved, CS-public, CDR3 sequences recognize similar antigenic epitopes that are conserved across species. These antigens may be derived from evolutionarily conserved regions of self proteins, forming a core of T cell reactivities to specific self epitopes, with potential implications for self-maintenance, autoimmunity and cancer. Further studies relating TCR$\alpha$, TCR$\beta$ and peptide specificity will enable to experimentally test this hypothesis.

Our results indicate that T lymphocytes 'focus their attention' to specific regions in sequence space. These new findings on the organization of TCR repertoires and their dynamics raise intriguing questions, for example, does the existence of network clusters indicate a healthy immune state? Can restoration of network structure reinstate immune function in the elderly or prevent excess inflammation and autoimmune disease? The theory of the immunological homunculus composed of self-recognizing B cells and T cells (*Cohen, 1992*, *2000*) might be relevant here.

## Materials and methods

### Mice

Female 5–8 weeks old C57BL/6 mice were obtained from Harlan Laboratories. Analysis of TCR sequences from aged mice is based on data that was previously described in *Shifrut et al. (2013)*. Analysis of TCR sequences from repertoires which are not subject to MHC-dependent selection, is based on Quad-KO mice, which are lacking four elements needed for physiological MHC-dependent antigen selection: MHC-I and -II molecules together with CD4 and CD8 co-receptor molecules, and matched control WT mice (*Van Laethem et al., 2007*, *2013*) and DN thymocytes, which represent the landscape of generated TCRs before thymic selection.

### Human data used in this study

Dataset of 39 healthy Caucasian donors, ages 6–90 years, was obtained from *Britanova et al. (2014)* (*Robert et al., 2014*). CTLA4 blockade data was obtained from *Robert et al. (2014)*. Juvenile Idiopathic Arthritis (JIA) data of patients compared to healthy donors was obtained from *Henderson et al. (2016)*.

### Immunization and in vitro stimulation

Mice were injected intra-peritonealy (IP) with 100 µg of either Chicken Ovalbumin (OVA) or peptide 277 (p277) emulsified in CFA (1:1 ratio). Spleens were harvested on day 7 post immunization and T cells were extracted for TCR analysis. in vitro stimulation: T cells from spleens of immunized mice were harvested on day 7 and were re-stimulated with irradiated splenocytes and the relevant peptide antigen. Five of the OVA-immunized mice received a boost IP injection of 100 µg OVA + CFA on day 14, and spleens were harvested on day 60 for TCR analysis (*Supplementary file 3*).

### Library preparation for TCR-seq and data pre-processing

Libraries were prepared and pre-processed as published (*Ndifon et al., 2012*). Briefly, T cells were purified from splenocytes by magnetic bead separation, total RNA was extracted and reverse transcribed using a TCR C$\beta$-specific primer linked to the 3'-end Illumina sequencing adapter. cDNA was amplified using PCR with a C$\beta-$3'adpater primer and a set of 20 V$\beta$-specific 5' primers, followed by ligation of a 5'Illumina adaptor and a second PCR using universal primers for the 5' and 3' Illumina adapters. The libraries were sequenced using Genome Analyzer II or HiSeq 2000 (Illumina). Sequence filtering, VDJ annotation, normalization and translation to AA sequences were performed as published (*Ndifon et al., 2012*). Libraries for TCR-seq of Quad mice and C57BL/6 controls were sequenced using Illumina sequencers, performed by Adaptive Biotechnologies Corp (Seattle, WA). In brief, $\alpha\beta$T cells were isolated by cell sorting, washed in PBS and lysed in Trizol. RNA was extracted using the RNEasy protocol (Qiagen) and 2 µg per sample reverse transcribed to cDNA by oligo (dT) priming with the SuperScript TM III First-Strand Synthesis System (Invitrogen). cDNA was sequenced by Adaptive Biotechnologies Corp.

## Statistical analysis and visualization

Statistical analysis was performed using R Software (*Core Team, 2013*). We used the following packages: 'ShortRead' (*Morgan et al., 2009*) for the pre-processing pipeline; 'ineq' (*Zeileis, 2012*) and 'reldist' (*Handcock, 2014*) to calculate the Gini coefficient; 'Igraph' (*Csardi and Nepusz, 2006*) to create network objects, obtain the degree of a node and its betweeness; 'stringdist' (*van der Loo, 2014*) to calculate Levenshtein distances; and 'ggplot2' (*Wickham, 2009*) for generating figures. Statistical tests performed are stated in the text. All network figures were made using Cytoscape (http://www.cytoscape.org/) (*Cline et al., 2007*; *Smoot et al., 2011*; *Saito et al., 2012*).

## Data access

The sequence data from this study have been made publicly available (https://usegalaxy.org/u/erezgrn/h/network-tcrs).

# Acknowledgements

We thank Benjamin Chain and Shalev Itzkovitz for helpful comments on the manuscript. This research was supported by grants from the Minerva Foundation with funding from the Federal German Ministry for Education and Research and the I-CORE Program of the Planning and Budgeting Committee and the Israel Science Foundation. AM was supported by the MD Moross Institute for Cancer Research.

# Additional information

## Funding

| Funder | Grant reference number | Author |
| --- | --- | --- |
| M.D. Moross Institute for Cancer Reseach | | Asaf Madi |
| Minerva Foundation | Funding from the Federal German Ministry for Education and Research | Nir Friedman |
| I-CORE | Program of the Planning and Budgeting Committee and the Israel Science Foundation | Nir Friedman |

The funders had no role in study design, data collection and interpretation, or the decision to submit the work for publication.

## Author contributions

AM, AP, Conceptualization, Formal analysis, Investigation, Writing—original draft; ES, Data curation, Formal analysis, Investigation, Writing—review and editing; SR-Z, Investigation, Methodology, Writing—review and editing; EG, TA, Formal analysis, Investigation; IZ, Investigation, Methodology; FVL, Resources, Formal analysis, Investigation; AS, Resources, Supervision, Investigation, Methodology; JL, Resources, Investigation, Methodology; PDS, Resources, Formal analysis, Investigation, Methodology; IRC, Conceptualization, Supervision, Writing—original draft; NF, Conceptualization, Supervision, Funding acquisition, Writing—original draft

## Author ORCIDs

Asaf Madi, http://orcid.org/0000-0003-3441-3228
Asaf Poran, http://orcid.org/0000-0001-9118-1051
Irena Zaretsky, http://orcid.org/0000-0003-4161-4677
Nir Friedman, http://orcid.org/0000-0002-1078-3921

## Ethics

Animal experimentation: This study was performed in strict accordance with the recommendations in the Guide for the Care and Use of Laboratory Animals of the National Institutes of Health. All of

the animals were handled according to approved institutional animal care and use committee (IACUC) protocols (#24110116-2) of the Weizmann Institute of Science. The protocol was approved by the Committee on the Ethics of Animal Experiments of the Weizmann Institute of Science. Every effort was made to minimize suffering.

## Additional files

### Supplementary files

• Supplementary file 1. Statistics of TCR networks for mouse and human repertoires. Mouse data: 12 mice from (*Madi et al., 2014*). Human data: 11 young subjects from (*Britanova et al., 2014*).

• Supplementary file 2. Summary of the data for the quad-KO mice, which are lacking four elements needed for physiological MHC-dependent antigen selection: MHC-I and -II molecules together with CD4 and CD8 co-receptor molecules (*Van Laethem et al., 2007*, *2013*), and matched control WT mice. Connected.nodes and edges refers to network statistics generated from the 1000 most frequent CDR3 sequences in each mouse.

• Supplementary file 3. Summary of TCR-seq data used in this study, from 5 experimental conditions: (1) mice that were immunized with either Chicken Ovalbumin (OVA) or (2) peptide 277 (p277), of HSP60. Spleens were harvested on day 7 post immunization and T cells were extracted for TCR analysis. (3) in vitro stimulation: T cells from spleens of immunized mice were harvested on day 7 and were re-stimulated with irradiated splenocytes and the relevant peptide antigen. (4) Five of the OVA-immunized mice received a boost IP injection of 100 μg OVA + CFA on day 14, and spleens were harvested on day 60 for TCR analysis. (5) DN thymocytes.

### Major datasets

The following previously published dataset was used:

| Author(s) | Year | Dataset title | Dataset URL | Database, license, and accessibility information |
|---|---|---|---|---|
| Friedman N | 2015 | Young mice TCR repertoire | https://www.ncbi.nlm.nih.gov/sra/SRP042610 | Publicly available at NCBI Sequence Read Archive (accession no: SRP042610) |

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
