## [Decision Letter]

Thank you for submitting your article "T cell receptor repertoires of mice and humans are clustered in similarity networks around conserved public sequences" for consideration by *eLife*. Your article has been reviewed by three peer reviewers, and the evaluation has been overseen by Arup Chakraborty as the Reviewing Editor and Senior Editor. The reviewers have opted to remain anonymous.

The reviewers have discussed the reviews with one another and the Reviewing Editor has drafted this decision to help you prepare a revised submission.

Summary:

The paper presents insightful analyses of T cell receptor sequence repertoires. Network analyses is used to identify clusters of CDR3β sequences, which are over-represented in peripheral T cell repertoires and are often found in multiple individual mice or humans (so called public TCRs). The analysis primarily concerns sequence data that was collected by the authors from a few distinct strains of mice (inbred wild type, plus a couple of immune system knockout strains). In addition, published human repertoire data is used to explore cross-species relevance of results obtained by analyzing mouse data. The authors focus on the most abundant sequences in the repertoire derived from individual animals (roughly the top.3 percent). They then ask whether these abundant sequences are distinguished by any global characteristics.

The authors find that the abundant sequences can largely be decomposed into subsets such that every sequence in a subset is connected to some other sequence in the same subset by at most one substitution, insertion or deletion. The second observation is that the large clusters contain (or have sequences that are one substitution/insertion/deletion away from) members of a group of 124 TCR sequences that have previously been annotated as responders to specific, identifiable, antigens. Interestingly, and enigmatically, the annotated TCR sequences that connect to these clusters mostly have annotations associated with self-reactivity. The authors develop these basic observation in several directions: 1) they show that a group of sequences selected for being present in multiple individual mice (25 out of 28) have similar properties of clustering and association with known antigens; 2) they show that similar clustering and antigen association of abundant TCR sequences occurs in human data and that there is strong overlap between abundant sequences in the two species; 3) they perform related analyses on mice that are knock-out with respect to various elements of the adaptive immune system and show that the sequence cluster organization of abundant sequences is not present if T cell activation is not possible (by knocking out the antigen presenting MHC complex, for example). In other words, that the sequence cluster organization is a product of T cell activation and response. 4) The authors go one to show that T cell selection provides a competitive advantage for selecting T cells that carry the more frequent TCR CDR3 sequences, i.e., T cell selection limits diversity by selecting against thymocytes expressing low frequency, highly variable CDR3bs. In addition, the authors provide evidence that T cell primary responses, CTLA4 checkpoint blockade and aging disrupts the "normal" TCR CDR3b frequencies; i.e., immune response diversifies the hierarchy of T cell CDR3b sequence frequencies presumably by expanding low-frequency T cells specific for particular antigens. The authors speculate that these frequency networks are reflective or perhaps required for proper T cell immune homeostasis.

While the paper is interesting, a number of points need to be addressed.

Major points to be addressed:

1) There is an over-emphasis on describing the network with minimal provision of primary data; e.g., it would be helpful to provide the actual sequences of each node.

2) Given the subject matter, there is a general lack of discussion regarding V-D-J recombination. Specifically, the following points need to be clarified:

Preface: human and mouse Db1 and Jb2.7 gene are 100% homologous (Db1 nucleotide, Jb2.7 AA sequence). Human Jb2.3 is highly similar to mouse Jb2.5, identical if 2 AA of the Jb are "chewed back”, which is relatively common during V-D-J rearrangement. Because of this sequence homology, CDR3s made from template-only V-D-J recombination using many Vbs, Db1 and Jb2.7 will by definition be identical in mouse and human. It stands to reason that insertions/deletions of these gene segments during recombination will also generate the identical sequences at a reasonable frequency.

"We discovered an unexpected number of public CDR3-TCRβ segments that were identical in mice and humans." Is this more so than would be expected given the extensive sequence homology between mouse and human Db/Jbs?

"These findings propose that similar driving forces may generate and expand particular public CDR3 TCR sequences that contain conserved sequence motifs in the two species." Given that template only V-D-J recombination of Db1 and Jb2.7 (or Jb2.3) would give identical TCRb CDR3 sequences, isn't sequence homology the evolutionary basis of public CS CDR3s?

3) Clarity of discussion of how CDR3β sequences relate to antigen specificity of a TCR. This evidence needs to be spelled out a bit in the main text. The reader is referred to other papers, but the point is so important that it would be appropriate to have a self-contained summary exposition in the paper itself.

4) Given that several aspects of novelty that the authors are claiming are known in other context or are predictable, the authors should directly test their hypothesis that disrupted TCR CDR3β networks are at the minimum a "biomarker" for the disease state; e.g., are there TCR CDR3β network signatures of chronic infection? There are several mouse models (LCMV, TB etc.) or human conditions that could be used as source material.

5) This point concerns Figure 3 and the discussion around it. Why all the nodes in each cluster are colored in the same way is not clear. Only a few nodes in a given cluster are identical to, or one step away from, one of the 124 annotated TCR sequences. Is the implication of the color scheme that any node in the cluster is expected to be responsive to one of the antigens that are identical (or close to) at least one node in the cluster? The discussion of this point not entirely clear.

6) In Figure 4 and the surrounding discussion, mention is made of network analysis of repertoires obtained from DN (double negative CD4- DC8-) thymocytes. This data set is not mentioned in Materials and methods, nor is any link to a repository provided. These data are extremely important as they bear on the question whether the highly shared TCR sequences are abundant because of antigen reaction and clonal expansion or due to some other cause. More detailed information about this data set should be given (how many sequences per DN mouse etc.) and, ideally, a pointer to the repository of this data should be given. The data repository should give the nucleotide sequences and not just the amino acid sequences of the CDR3 since the text makes a point of the difference in the number of nt realizations of specific CDR3 aa sequences when comparing the DN mice with the WT mice.

---

## [Author Response]

Major points to be addressed:

*1) There is an over-emphasis on describing the network with minimal provision of primary data; e.g., it would be helpful to provide the actual sequences of each node.*

As it is visually problematic to present the entire network with the actual sequences, we added to Figure 1 in the main text an inset panel that captures one of the main clusters with the AA sequences for each node presented. This example demonstrates the construction method of these networks.

*2) Given the subject matter, there is a general lack of discussion regarding V-D-J recombination. Specifically, the following points need to be clarified:*

*Preface: human and mouse Db1 and Jb2.7 gene are 100% homologous (Db1 nucleotide, Jb2.7 AA sequence). Human Jb2.3 is highly similar to mouse Jb2.5, identical if 2 AA of the Jb are "chewed back”, which is relatively common during V-D-J rearrangement. Because of this sequence homology, CDR3s made from template-only V-D-J recombination using many Vbs, Db1 and Jb2.7 will by definition be identical in mouse and human. It stands to reason that insertions/deletions of these gene segments during recombination will also generate the identical sequences at a reasonable frequency.*

*"We discovered an unexpected number of public CDR3-TCRβ segments that were identical in mice and humans." Is this more so than would be expected given the extensive sequence homology between mouse and human Db/Jbs?*

We thank the reviewers for raising this point, which made us refine our statement. In order to answer this question we generated simulated TCR repertoires using V-D-J genes of either mouse or human. Other parameters of the simulations (for example, frequencies of usage of the V-D-J genes, frequencies of random nucleotide insertions/deletions) were determined from data (using “unselected” TCR sequences from the data – those that have a stop codon or are out of frame – about 1-2% of the sequences). We used a similar method in a previous publication: Madi et. al, Genome Research 2014.

We generated 100 datasets of simulated repertoires of 28 mice, the sizes of the repertoires match the sizes of the experimental 28 murine repertoires. For each of these 100 datasets, mouse simulated public sequences were defined as sequences that appear in 25 simulated repertoires or more, as we did for the experimental data). Similarly, we generated 100 datasets of simulated repertoires of 11 human each (simulation parameters were extracted from the data by Britanova et. al 2014, from which we also used the data in the original analysis). Human simulated public sequences were defined as those that appear in all 11 human repertoires in a simulated dataset, as done for the experimental data. Finally, simulated cross-species public sequences were defined based on the overlap between the simulated datasets.

We then tested these cross-species public sequences for J segment usage. Most of these sequences indeed had the mouse Jb2.7 or 2.5, and the human 2.7 or 2.3. Thus, sequence homology between these J segments does contribute to the generation of CS-public sequences. However, this by itself is not a sufficient condition, as there are many sequences that have these J segments, and are public in one species but not in the other (both in the simulations and in the experimental data).

We note that in these simulations, cross-species public sequences were generated at a somewhat higher number than in the experimental data: 221 CS-public on average in the simulations, compared with 86 CS-public in the experimental data. Thus, we don’t use “unexpected” to describe the number of CS-public sequences found. We are aware that these simulations somewhat overestimate the sharing between individual repertoires in both species (as noted in our Genome Research paper). This may stem from inaccuracies in the simulation assumptions, or due to the fact that we simulate only the random generation process of TCRs, but not their selection (in the thymus and in the periphery).

As for the actual sequences, of the 86 CS-public sequences in the data, only 54 are CS-public also in the simulations. The other 32 are found in the simulations at lower sharing levels. This is presented in Figure 2—figure supplement 2. Combined, these results show that the simulations predict the existence of CS-public TCR CDR3β sequences, but not all sequences that are observed as CS-public are also CS-public in the simulation, and vice versa. This suggests that conserved selective forces shape and refine the likelihood of a CDR3β sequence to be CS-public.

To conclude, this analysis suggests that sequence homology together with the parameters of the random generation process drive the generation of CS clones; however their existence is further pruned and shaped by selective forces that are absent from the simulation.

We added in the revised manuscript description of the simulations and their results, and revised the discussion of CS-public clones accordingly.

*"These findings propose that similar driving forces may generate and expand particular public CDR3 TCR sequences that contain conserved sequence motifs in the two species." Given that template only V-D-J recombination of Db1 and Jb2.7 (or Jb2.3) would give identical TCRb CDR3 sequences, isn't sequence homology the evolutionary basis of public CS CDR3s?*

As a further test for other forces that influence the level of CS-public sequences, we compared the abundance of CS-public sequences in repertoires of the quad-KO mice, which are not subject to MHC-dependent selection, to those of control WT mice. We made a similar comparison also with repertoires of DN thymocytes, which represent the landscape of generated TCRs before thymic selection. The cumulative frequencies of the CS CDR3 sequences between two sets of experiments done with WT mice (the 28 WT mice used in the network analysis, and the WT mice used as controls in the quad-KO experiment) show no significant difference (P value = 0.293). On the other hand, the QuadKO exhibited lower total frequency of the CS CDR3s compared with both 28 WT (P value = 4.318e-09) and the QuadWT (P value = 0.01781). The cumulative frequency in the DN shows a similar trend, with no statistical significant (P value = 0.1877). Together, these results indicate that although sequence homology defines a probability for a sequence to be shared between the two species, additional selection forces are influencing its abundance.

These results were now added to the main text, and as a new panel in Figure 4.

*3) Clarity of discussion of how CDR3β sequences relate to antigen specificity of a TCR. This evidence needs to be spelled out a bit in the main text. The reader is referred to other papers, but the point is so important that it would be appropriate to have a self-contained summary exposition in the paper itself.*

To expand the discussion of how CDR3β sequences relate to antigen specificity, we added the following to the text, when describing Figure 3: “The functional TCR is formed by a complex of TCR α and β chains (Davis and Bjorkman 1988), hence one cannot attribute specific antigen recognition to CDR3β segments alone. […] Some insight on antigen specificity can be gained by linking the sequence-similarity networks to previously annotated TCR sequences.”

*4) Given that several aspects of novelty that the authors are claiming are known in other context or are predictable, the authors should directly test their hypothesis that disrupted TCR CDR3β networks are at the minimum a "biomarker" for the disease state; e.g., are there TCR CDR3β network signatures of chronic infection? There are several mouse models (LCMV, TB etc.) or human conditions that could be used as source material.*

The data presented in the manuscript, specifically in Figure 5, introduce various perturbations which are associated with network deterioration. First, we show that exposure and re-exposure of the repertoire to an antigen via immunization leads to network deterioration, due to preferential expansion of relatively private clones at the expense of the more highly connected public clones that support the network structure. This was observed for immunization with a foreign antigen (OVA) and also with a self-antigen (HSP60). We also show that recovery following exposure in the OVA immunized mice restored a naive-like network structure. Most of this analysis in the paper is based on new experiments that we conducted, which were not described in our previous publications.

We further show that network connectivity in human patients was significantly reduced following the strong perturbation associated with immune checkpoint blockade. Lastly, we show a strong correlation between aging, associated with decreased immune functioning, and network connectivity. Together we conclude that network strength is a proxy for immune state.

Following the reviewers’ suggestion, we searched for examples of network structure and sharing modifications in other diseases. We added a new analysis to the revised manuscript, not included in the original manuscript, of patients with the autoimmune disease Juvenile Idiopathic Arthritis (JIA) (Henderson et al. 2016). We found that there was a strong increase of public (network promoting) TCRs in the peripheral blood of JIA patients compared to healthy donors (P value = 0.0006, see figure below). This finding shows that while immune perturbations such as immunization and aging lead to expansion of private clones and network reduction, this specific autoimmune condition is associated with an increased level of public clones which are putatively associated with self-antigens. Taken together, our analysis supports the idea that the level of network connectivity, frequency of public TCRs and repertoire evenness or skewing are linked to each other, and are concurrently modulated by immune state – disease / immunization / aging. However, given the magnitude of the observed effect of disease on network structure, larger datasets of TCR repertoires of sick vs. healthy people are required in order to define network-based features as biomarkers. Further work is required to indicate the conditions / diseases that modulate network structure more than others, and also maybe to refine the populations of T cells used for network generation (e.g. build networks of effector, or memory T cell TCRs). These directions are under investigation by us, beyond the scope of the current manuscript.

We have added to the revised version a panel (Figure 5), text that describes these results, and also refer to them in the Discussion.

*5) This point concerns Figure 3 and the discussion around it. Why all the nodes in each cluster are colored in the same way is not clear. Only a few nodes in a given cluster are identical to, or one step away from, one of the 124 annotated TCR sequences. Is the implication of the color scheme that any node in the cluster is expected to be responsive to one of the antigens that are identical (or close to) at least one node in the cluster? The discussion of this point not entirely clear.*

We thank the reviewers for noting this. The color of the clusters was meant only as a way of distinguishing between clusters in the figure, not to define antigen specificity, which is now further clarified in the figure legend. We note (and explain in the text, including specific examples) that clusters can include sequences of different annotations. Thus, clusters are not homogenous in terms of antigen specificity, despite the high level of sequence similarity. Some nodes, in particular close ones, may share the same antigen (as indicated by annotated clonotypes in some cases), but this is not general. This issue is of course further complicated by TCR cross-reactivity. Also, as we note in the text, TCR specificity can depend also on the TCRa part of the receptor. To clarify our notation, we have added a supplemental figure (Figure 3—figure supplement 1), in which we specifically label the nodes (CDR3 sequences) in one large cluster, and identify those that are annotated. We also improved the presentation of annotated colontypes in Figure 3, and revised the text that discusses this figure.

*6) In Figure 4 and the surrounding discussion, mention is made of network analysis of repertoires obtained from DN (double negative CD4- DC8-) thymocytes. This data set is not mentioned in Materials and methods, nor is any link to a repository provided. These data are extremely important as they bear on the question whether the highly shared TCR sequences are abundant because of antigen reaction and clonal expansion or due to some other cause. More detailed information about this data set should be given (how many sequences per DN mouse etc.) and, ideally, a pointer to the repository of this data should be given. The data repository should give the nucleotide sequences and not just the amino acid sequences of the CDR3 since the text makes a point of the difference in the number of nt realizations of specific CDR3 aa sequences when comparing the DN mice with the WT mice.*

We regret this omission in our submission. We have added description of this data to the Materials and methods section, added tables that describe these datasets (as well as the quad-KO data, which was also missing) to the supplementary information (no. of sequences per mouse, etc.,) ([Supplementary-material SD2-data] and [Supplementary-material SD3-data]), and also uploaded the nt data to a public repository (https://usegalaxy.org/u/erezgrn/h/network-tcrs).